# Association between Cardiovascular Risk and Diabetes with Colorectal Neoplasia: A Site-Specific Analysis

**DOI:** 10.3390/jcm7120484

**Published:** 2018-11-27

**Authors:** David Niederseer, Isabelle Bracher, Andreas Stadlmayr, Ursula Huber-Schönauer, Martin Plöderl, Slayman Obeid, Christian Schmied, Sabrina Hammerl, Felix Stickel, Dieter Lederer, Wolfgang Patsch, Elmar Aigner, Christian Datz

**Affiliations:** 1Department of Internal Medicine, General Hospital Oberndorf, Teaching Hospital of the Paracelsus Medical University Salzburg, 5110 Oberndorf, Austria; david.niederseer@gmx.at (D.N.); andreas.stadlmayr@schur.at (A.S.); huber.schoenauer@gmail.com (U.H.-S.); sabrinahammerl@yahoo.com (S.H.); d.lederer@kh-oberndorf.at (D.L.); 2Department of Cardiology, University Heart Centre, University Hospital Zurich, 8091 Zürich, Switzerland; isabelle.bracher@gmail.com (I.B.); slayman.obeid@usz.ch (S.O.); christian.schmied@usz.ch (C.S.); 3Suicide Prevention Research Program, Paracelsus Medical University, 5020 Salzburg, Austria; m.ploederl@salk.at; 4Department of Suicide Prevention, University Clinic of Psychiatry and Psychotherapy I, Christian Doppler Clinic, 5020 Salzburg, Austria; 5University Institute of Clinical Psychology, Christian Doppler Clinic, 5020 Salzburg, Austria; 6Department of Gastroenterology and Hepatology, University Hospital of Zürich, 8091 Zürich, Switzerland; felix.stickel@uzh.ch; 7Institute of Pharmacology and Toxicology, Paracelsus Medical University Salzburg, 5020 Austria; wolfgang.patsch@pmu.ac.at; 8Department of Internal Medicine I, Paracelsus Medical University Salzburg, 5020 Salzburg, Austria; e.aigner@salk.at; 9Obesity Research Group, Paracelsus Medical University Salzburg, 5020 Salzburg, Austria

**Keywords:** Framingham Risk Score (FRS), Heart Score (HS), site-specific colorectal neoplasia, cardiovascular risk

## Abstract

Several studies have shown site-specific differences in colorectal cancer (CRC) with respect to the risk factors. CRC was shown to be associated with cardiovascular risk (CVR) factors, but site-specific variations have not been investigated so far. This study aimed to assess the associations between the prevalence and subsite-specific differences of colorectal neoplasia and established CVR scores or known coronary artery disease (CAD) in a large asymptomatic European screening cohort (*N* = 2098). Participants underwent simultaneous screening colonoscopy and CVR evaluation, using the Framingham Risk Score and Heart Score. Lesions found in the colonoscopy were classified by location (proximal/distal colon or rectum). More neoplasias were found in the proximal versus the distal colon (*p* < 0.001). The Framingham Risk Score and Heart Score showed incremental risk for colorectal adenoma, across the tertiles in the proximal and the distal colon (*p* < 0.001). The prevalence of adenomas in the rectum was much lower, but also here, incremental risk could be shown for the Framingham Risk but not the Heart Risk Score tertiles. Prevalence of adenomas in the proximal colon was higher in subjects with type 2 diabetes (T2DM) (*p* = 0.006), but no association was found between adenomas and T2DM in the distal colon (*p* = 0.618) and the rectum (*p* = 0.071). Males had a higher CVR and more findings, in the screening colonoscopy, as compared to females, however, no site-specific differences were noted. Patients with known CAD and high CVR have an increased risk of colorectal neoplasia in both the proximal and distal colon. Patients with T2DM have a higher risk for neoplasia in the proximal colon.

## 1. Introduction

Colorectal cancer is one of the most common cancers worldwide. In industrialized countries, it is the third most common cancer in men, after those of the lung and the prostate, and the second most common in women, after breast cancer [1]. 

Several differences between the proximal colon (i.e., from the caecum to the transverse colon), the distal colon (i.e., from the distal colon flexure to the rectum), and the rectum can be found; including their embryological origin, histological features, physiological function, fecal composition, intestinal transit time and even the mechanisms of oncogenesis [2,3,4,5]. In 1990, Bufill was the first to propose the concept that cancers originating from the proximal and the distal colon may be viewed as two distinct forms of cancer, with rectal cancer being a third cancer subtype, all of which are usually summarized as colorectal cancer (CRC) [2,6]. More recent studies demonstrating the clinical, histopathological, and molecular differences have supported this theory [7].

Furthermore, several studies have been conducted to assess the subsite-specific risk factor profiles for cancers in different anatomical locations of the colon and the rectum, with inconclusive results [8,9,10,11,12,13,14].

Emerging evidence suggests an association between coronary artery disease (CAD) and CRC, possibly due to the shared risk factors, such as obesity, sedentary lifestyle, diabetes mellitus, hypertension, and cigarette smoking [15,16,17,18,19,20,21]. In a recently published study we showed that colorectal cancer was associated with cardiovascular risk factors in a large Caucasian cohort [22]. To the best of our knowledge, we are not aware of studies investigating if the associations between cardiovascular risk factors differ between the different sites of colorectal neoplasias, in screening colonoscopy. Thus, the aim of this study was to assess the site-specific differences of colorectal neoplasia in patients with differing cardiovascular risk profiles.

## 2. Methods

### 2.1. Subjects

The study was conducted using participants of a health screening program (Salzburg Colon Cancer Prevention Initiative, SAKKOPI) at the Department of Internal Medicine, Oberndorf Hospital, between 2010 and 2014. All patients without gastrointestinal symptoms that wished to be screened according to the national screening recommendations and agreed to participate in the study, were included [23,24]. This study complies with the Declaration of Helsinki and was approved by the local ethics committee (Ethikkommision des Landes Salzburg, approval no. 415-E/1262/2-2010). Informed consent was obtained from all participants.

### 2.2. Clinical Assessment

In this single-centre study, study participants were examined on two consecutive days. On the first day, the participants filled out a standardized questionnaire, underwent a physical examination, and venous blood was collected, after an overnight fast. On the second day, a screening colonoscopy was performed.

In total, two thousand one hundred and thirty-eight Caucasian subjects, without gastrointestinal symptoms, were screened. One hundred and twelve subjects presenting with gastrointestinal symptoms were excluded from the study.

Blood pressure was measured twice after a 5 min rest in a sitting position and the average was taken, arterial hypertension was defined as ≥140/90 mmHg. Waist circumference was measured at the highest point of the iliac crest, with subjects standing in an upright position. Metabolic syndrome was evaluated as defined by the National Cholesterol Education Program Adult Treatment Panel [25]. Body mass index was calculated as weight divided by squared body height (kg/m^2^).

### 2.3. Questionnaire

A detailed medical history, including the participant’s current medication, was obtained. Smoking status was classified as “never smokers”, “former smokers”, and “current smokers”. A reported history of coronary artery disease was verified by a review of the medical records and it was checked whether the diagnosis had been established by coronary angiography.

### 2.4. Laboratory Assessment

Cholesterol, triglycerides, HDL (high density lipoprotein) cholesterol, insulin, glucose, and HbA1c (glycated hemoglobin A1c) were measured in the fasting venous blood probe. A standardized oral glucose tolerance test was performed with 75 g of glucose in 300 mL of water. Type 2 diabetes was classified as the use of diabetes medication, HbA1c ≥ 6.5%, oral glucose tolerance test > 11.1 mmol/L, after 2 h, or fasting glucose >7.0 mmol/L, respectively. Impaired fasting glucose (IFG) was defined as fasting glucose <7.0 but >6.1 mmol/L, impaired glucose tolerance (IGT) was defined as oral glucose tolerance test <11.1 but >7.8 mmol/L, after 2 h [26]. Homeostatic Model Assessment for Insulin Resistance (HOMA-IR) was calculated, as previously suggested, to estimate the insulin resistance: HOMA-IR = (Fasting insulin [mg/dL]) ∗ (Fasting glucose [mg/dL])/405 [27].

### 2.5. Cardiovascular Risk Assessment

Two widely used cardiovascular risk scores were calculated for every subject—the Framingham Risk Score (FRS) and the Heart Score (HS), for low-risk countries (as Austria is a low-risk country) of the European Society of Cardiology (ESC) [28,29]. Forty subjects aged >79 years had to be excluded from the study, as the FRS was not validated in these patients. HS was originally designed for subjects aged 40–65 years, however, the formula allows to calculate the HS for subjects >65 years with the caveat that the cardiovascular risk will most likely be underestimated in these subjects. Both FRS and HS are not validated in diabetic patients. We did, however, include diabetic subjects in our analysis and performed a separate analysis, excluding all diabetic subjects. As the results of including or excluding diabetics were essentially the same, we report the results including the diabetic subjects to allow for greater generalizability of our results.

### 2.6. Colonoscopy

The laxative Polyethylene Glycol and Electrolytes (Klean-Prep^®^) was used for bowel preparation. After analysis of the macroscopic and the histologic results, the colonoscopic findings were classified as tubular adenoma, advanced neoplasia, i.e., adenoma with villous or tubulovillous features, size ≥1 cm or high-grade dysplasia, or carcinoma [30,31]. Lesions were classified by location—the proximal colon (including caecum), the ascending and transverse colon, the distal colon ranging from the splenic flexure to the sigmoid, and the rectum [32].

### 2.7. Statistics

We used Statistica 7.0 (Statsoft) and Stata 13.0 (Stata Corporation, College Station, TX, USA) statistical software for the calculations. For comparison of the categorical variables, we used a contingency χ^2^ test and the non-parametric nptrend test, across the ordered groups. We estimated the odds ratios (ORs), with 95% confidence intervals (CI), by univariate logistic regression analysis. To provide separate ORs for the middle and the upper FRS and the HS tertiles, we used two “dummy” variables with the respective low-tertile as a reference. Similar analyses were used to estimate the ORs for low, intermediate, and high CVR groups, stratified according to the various FRS and HS levels that have been previously used by others [33,34].

To translate the results into a clinically-relevant effect size, we report the number needed to be screened (NNS), i.e., the total number of subjects screened divided by the number of subjects with a pathological finding in the screening colonoscopy. The calculation of the AUC (area under the curve) was realized with R’s “pROC“ package (receiver-operating-characteristic), based on the continuous measures of the FRS and HS. Sensitivity (true positives/(true positives + false negatives)) and specificity ((true negatives/(true negatives + false positives)) for being in the high cardiovascular risk tertile was likewise calculated. Correlation matrix with Spearman’s rho was calculated to test for correlation of pre-diabetic states with the findings of the screening colonoscopy.

## 3. Results

### 3.1. Differences by Coronary Risk Profiles

Two thousand and ninety-eight subjects were included in the final analysis, including one hundred and eight subjects (5%) with a self-reported history of CAD, which was verified using the health records. The proportion of men, the mean values for age, BMI (body mass index), and FRS and HS scores, were significantly higher in subjects with a history of CAD, in comparison to the subjects without CAD history. Moreover, type 2 diabetes mellitus and metabolic syndrome were significantly more common in the group with a CAD history, while the prevalence of hypertension, measured blood pressure, and smoking history were comparable in the two groups (see Table 1). Except for Aspirin use, the clinical characteristics (shown in Table 1) did not differ significantly between the angiographically-verified CAD group and the group without angiography but with a positive CAD history (Appendix A).

Subjects without a history of CAD were divided into the approximate tertiles, according to their FRS and HS. The FRS upper limits of 3% and 8%, for the low- and intermediate-risk groups, were used. Thus, seven hundred and eleven subjects were assigned to the low, six hundred and thirty-three to the intermediate, and six hundred and forty-six to the high tertiles. The same tertile sizes were used for the HS, to facilitate comparison.

### 3.2. Cardiovascular Risk Profile and Site-Specific Colonoscopic Results

Subjects with known CAD history and high cardiovascular risk profiles had a higher probability of colorectal neoplasia, as reported previously, see Figure 1, Table 2, Table 3 and Table 4 [22].

For the site-specific analysis, as the number of advanced neoplasia in our population was small, we were only able to determine associations with the presence of any adenoma. Generally, the proportion of lesions was greater in the proximal than the distal colon (*p* < 0.001), and lowest in the rectum. For both the proximal and the distal colon, the number of lesions gradually increased from the lowest to the highest tertile of the FRS and HS, with the ORs ranging from 1.7 to 3.0 (*p* < 0.001, with the first tertile as the baseline) (Figure 2a–d). For the rectal lesions, similar associations were observed for the FRS, however, for the HS, the proportions of the rectal lesions were comparable in the second and the third tertile. In summary, the FRS and the HS showed an incremental risk for colorectal adenoma across the tertiles, with the exception of HS, with regard to the rectal adenomas (Figure 2, Table 5 and Table 6). The sensitivity and specificity calculations of the site-specific adenoma showed that high cardiovascular risk, as estimated by the FRS or the HS, is not sensitive (48–51%), but is specific (68–71%, depending on location) for colorectal adenoma. Likewise, the ROC analysis revealed that values ranged from 60–65% for the FRS (proximal colon: 64%; distal colon 65%; rectum 62%) and the HS (proximal colon: 64%; distal colon 64%; rectum: 60%).

Previously, the FRS was used to distinguish low, intermediate, and high cardiovascular risk, by using <11%, 11–20%, and >20% as the cut-off values. Likewise, the cut-off values of HS were <1%, 1–5%, >5% or <3%, 3–6%, and >6%. Results for these strata are shown in Appendix A; they led to largely comparable results.

### 3.3. Diabetes Mellitus and the Metabolic Syndrome

Subjects diagnosed with metabolic syndrome had a significantly higher cardiovascular risk estimated both by the FRS tertiles (*p* < 0.001) and the HS tertiles (*p* < 0.001). Whereas, the prevalence of adenomas in the proximal colon was significantly higher in subjects with type 2 diabetes (*p* = 0.006), no statistically significant relationship between the adenomas and type 2 diabetes was observed in the distal colon (*p* = 0.618) and the rectum (0.071). As both the FRS and the HS were initially designed for non-diabetic subjects, we re-calculated our entire results for non-diabetic subjects only (i.e., 1990 − 262 = 1728), with essentially the same results (data not shown). As in a real-world setting of the screening colonoscopy, diabetic and non-diabetic subjects are screened alike, we refrained from excluding all diabetic subjects, a priori from our analysis, in order to reach a higher statistical power and a greater generalizability of our results. A correlation matrix including the IFG, IGT, and HOMA-IR is depicted in Table 7. The FRS and HS performed better than HOMA, IFG, or IGT. 

### 3.4. Comparison for Sex Differences

As male gender and age is a risk factor for both cardiovascular diseases and CRC, we performed a sex-specific analysis of our data, as shown in Table 8. There was no difference in age between males and females. Generally, males had an increased CVR, as compared to females, and accordingly a significantly higher number of findings, during the screening colonoscopy. There were, however, no site-specific differences of the detected lesions between the male and the female participants.

## 4. Discussion

We report that cardiovascular risk, regardless of using the FRS or the HS, was associated with an increased risk of adenomas in the proximal and the distal colon, in asymptomatic subjects undergoing screening colonoscopy. In the rectum, the difference in distribution was more pronounced in the FRS. Furthermore, in patients with type 2 diabetes, we found significantly more adenomas in the proximal colon.

### 4.1. The Rationale for Site-Specific Risk Factors in CRC

As previously suggested, several biological differences might be present among malignancies originating from the proximal and the distal colon and the rectum, which lead to the suggestion of splitting these entities into three different tumors [4]. These biological differences might explain why several risk factors might affect the CRC risk, differently, by location. First, the embryological differences among the different locations might affect the susceptibility to carcinogens. The area including the cecum, the ascending colon, and the proximal two-thirds of the transverse colon, originates from the midgut, while the rest originates from the hindgut [2,35]. Second, subsite differences may be due to differences in tumor methylation status, BRAF (gene encoding for b-raf i.e. *rapidly accelerated fibrosarcoma)* and KRAS (proto-oncogene that acts as an on/off switch in cell signaling, named after Kirsten RAt Sarcoma virus) mutation status, as well as DNA microsattelite instability. Proximal tumors appear to be related to the CpG island methylator phenotype (CIMP)-high and the BRAF-mutated tumors, while the distal tumors were found to be mostly related to the no-CIMP and microsatellite stable tumors. Additionally, tumors in the cecum were frequently associated to a high frequency of the KRAS mutations [36].

Several authors have studied the subsite-specific risk factors, with conflicting results [8,10,11,12,13,14]. A systematic review on the subject, by Benedix et al., concluded that proximal colon cancer was found more frequently in women, patients of older age, and in regions with lower incidence rates for colorectal carcinoma. Furthermore, they described a better response rate to 5-fluoruracil-based chemotherapy in patients with right-sided colon carcinoma [37]. Such site-specific differences would have several clinical implications. On one hand, patients at risk for proximal colon cancer might benefit especially from a complete colonoscopy. On the other hand, future genetic, clinical, and treatment studies need to assess not only the effects of the medication in general, but also compare the effects in relation to the location of the tumor (i.e., proximal vs. distal colon). Should the hypothesis of two different tumor entities be confirmed, a location-specific therapy, and, in the case of subsite-specific risk factors, a location-specific screening and preventive strategy, according to the risk factors, may be required.

### 4.2. Site-Specific Risk Factors for CRC

Therefore, the main goal of this study was to investigate the site-specific differences in patients with different FRS and HS risk scores. FRS and HS showed incremental risk for colorectal adenoma, across tertiles, in both proximal and distal colon. However, in the rectum, the difference in distribution was more pronounced in the FRS. Along with a different prediction (FRS: ten-year risk of heart attack, HS: ten-year risk of cardiovascular death), the FRS includes the HDL-cholesterol, which the HS does not, whereas, the HS uses different models for different countries of origin. The more accurate association with FRS might underline the importance of the HDL-cholesterol, a risk factor that can be altered mainly by lifestyle interventions (healthy diet, regular exercise), in the rectal neoplasm. Howard et al. have previously shown that regular physical activity is more protective in the rectal than in the colonic neoplasm [12]. Interestingly, our data show the expected increased CVR in males and, consequently, a higher detection-rate of premalignant lesions, during the screening colonoscopy. However, we did not find sex differences with regard to the site-specific aspects, which other studies have found [37].

In line with our findings, in a community-based cohort of adults aged >50 years, Laird-Fick et al. showed that the polyps were most often found in the right sided colon, whereas the prevalence of polyps in the rectum was lowest with only 0.7% of lesions found in the rectum [38].

### 4.3. Diabetes Mellitus Type 2 and Metabolic Syndrome

Previously, the metabolic syndrome and type 2 diabetes have been linked to CRC [39]. In a meta-analysis by Larsson et al., it was concluded that diabetes mellitus was associated with an increased CRC risk in both men and women [40]. This association was also found in more recent meta-analyses [36,41]. Already in 1995, a hypothesis involving hyperinsulinemia was proposed to explain the association between diabetes mellitus and CRC risk [42]. It was presumed that the insulin resistance in patients with diabetes mellitus leads to a chronic hyperinsulinemic state and elevation of insulin-like growth factor-1 levels; this plays a crucial role in the cell proliferation and finally in the occurrence of CRC [42]. Furthermore, Kiunga et al. conducted an in vivo study in which they showed that insulin receptors contribute to the cell transformation [43]. This led to the hypothesis that elevated insulin receptor protein expression in colonic tumors might be a possible mechanism for tumorigenesis.

Regarding subsite-specific differences in patients with type 2 diabetes, several studies have stated conflicting results [10,12,44]. When studying site-specific differences in patients with type 2 diabetes, we found that they had significantly more adenomas in the proximal than in the distal colon. Our analysis revealed interesting results, in that, although the proportion of subjects with metabolic syndrome and type 2 diabetes was significantly higher in the highest tertiles of both the FRS and the HS, the metabolic syndrome and type 2 diabetes itself were not, or were only marginally associated with colorectal adenomas, with one exception—subjects with type 2 diabetes had significantly more adenomas in the proximal colon. This further highlights the need for a better understanding of the complex relationship between diabetes, the metabolic syndrome, and cancer. As FRS and HS perform better in our correlation matrix than the IFG, IGT, and the HOMA-IR, indicating an insulin resistance, this suggests the incremental importance of pre-diabetic states to diabetes, on the development of colorectal lesions.

### 4.4. Number Needed to Screen: Selecting High-Risk Patients for CRC to Increase the Detection-Rate

Several prior investigations studied various risk scores to select the high-risk subjects in a screening cohort and consequently lower the number needed to screen [45,46,47,48,49,50,51]. Variables that were considered included, age, sex, family history of CRC, BMI, smoking, dietary habits, prior colonoscopy and the results, thereof, included the use of non-steroidal anti-inflammatory drugs and waist circumference. The degree of lowering of the numbers needed to screen we report, herewith, is comparable to these previously published studies [45,46,47,48,49,50,51].

Since up to 10% of CRC occur in subjects younger than 50 years and also myocardial infarction is seen in 4–10% in subjects ≤ 40–45 years, we think there is no good reason to a priori exclude the younger subjects from the analysis, as others have done [52,53]. 

Several reports, with conflicting results related to the preventive nature of statins and metformin, have been published previously [54,55]. We, therefore, did not control our analysis for the use of statins or metformin. With respect to Aspirin, the published evidence strongly suggests a protective effect against CRC [56,57]. As most Aspirin users have an increased CVR, in our cohort, Aspirin was not protective but rather associated with a higher risk, presumably reflecting the higher prevalence of common risk factors of subjects taking aspirin. 

In light of the emerging importance of disease prevention and health screening programs, the previously published studies, as well as our data, underline the need for screening initiatives including screening colonoscopy. In health screening initiatives, in general, the NNS is crucial. With regard to screening colonoscopy in a large Austrian screening cohort, Ferlitsch et al. reported an NNS of 5.1 (95% CI, 5.0–5.2), 15.9 (95% CI, 15.4–16.5), and 90.9 (95% CI, 83.3–100.0), for colorectal adenoma, advanced adenoma, and CRC, respectively [58]. In representative studies on screening colonoscopy in different ethnicities, and the risk estimation in colonoscopy, the main outcome that was defined as advanced adenoma/neoplasia, varied between 3.0% and 11.2% [42,43,44,45,46,47,48]. Although the number of advanced neoplasia we report is at the lower end of this range (3.8%), our NNS was below the previously reported value by Ferlitsch et al., as only four subjects had to be screened to detect an adenoma, and only twenty seven to detect an advanced neoplasia (i.e., advanced adenoma and CRC). Ferlitsch et al. reported an advanced adenoma and CRC, separately; unfortunately, we could not provide meaningful calculations, due to the lower number of screened subjects in our study (44,350 vs. 2089) [58]. However, the numbers needed to screen, which we have reported, get even lower, as the CVR rises (in high CVR, for adenoma, it is three for advanced neoplasia, depending on the risk score—15 or 17), indicating a solid basis for a recommendation of screening colonoscopy, especially in subjects with high CVR. Of note, the NNS were lower in the proximal colon (6.1) than in the distal colon (8.1), and rectum (21.4), and the increment of the NNS, as the CVR changed, persisted in all three colorectal regions.

### 4.5. Coronary Artery Disease, Cardiovascular Risk, and Colorectal Cancer: Potential Mechanisms, Integration into a Holistic Diagnostic Approach, and Future (Practical) Considerations

The association between CAD, diabetes, and colorectal neoplasia that we have reported in this paper, raises questions about the possible common pathogenetic factors. Several possible explanations have previously been suggested, among them are the shared risk factors, such as obesity, sedentary lifestyle, diabetes mellitus, hypertension, and cigarette smoking [15,16,17,18,19,20,21]. On the molecular level, several shared risk factors may underlie the common pathways, with inflammation appearing to be a major contributor. However, insulin resistance, reactive oxygen species and oxidative stress, hormones (e.g., leptin), cytokines, growth factors, and other metabolic reactions have also been linked to CAD and cancer. [21]. Only recently, HMGA-1 (High-mobility group protein A1) a structural chromatin protein was shown to be involved in cancer and its invasiveness in breast cancer [59]. In addition, a specific variant of HMGA-1 has been found to increase the risk for diabetes, coronary artery disease, and acute myocardial infarction [60]. This fascinating interaction of the same protein being involved in cancer invasiveness and risk for diabetes, CAD, and acute coronary syndrome, offers a potential molecular explanation of our findings that, to our knowledge, has not been studied yet.

Also, the prognostic value of noninvasive biomarkers and investigations, such as microRNAs, microbiota-related aspects, or noninvasive imaging, are of interest, with regard to our data in further risk stratifying subjects, for screening colonoscopy. MicroRNAs have shown promising preliminary results for both diagnostic and prognostic assessment of cardiovascular disease in patients with diabetes and, if integrated in a holistic risk stratification tool, might help to select the appropriate subjects in the right time-frame, to avoid unnecessary screening procedures [61]. We have previously shown the value of using a specific pattern of microbiota to test stools of screening candidates [62]. It is now widely accepted that microbiota is also involved in atherosclerosis, however, the exact overlap of microbiota between cardiovascular and gastrointestinal pathologies remains to be elucidated. Noninvasive imaging biomarkers might also be helpful in this context. In particular, a novel modality to assess the flow-mediated dilation that takes into account the time needed for dilation of the brachial artery, was recently tested, and the results showed promise for the successful exclusion of the presence of critical coronary stenoses (>=70%), before performing invasive coronary angiography [63]. Again, data of tests, such as flow-mediated dilation could help to better recommend the time and population for screening colonoscopy, in terms of a personalized screening approach.

We report an association between colorectal neoplasia and CAD, meaning that no conclusions regarding causality can be drawn from our data. It follows naturally that the association works in both ways. Cardiologists or diabetologists that take care for patients with CAD, increased CVR, or diabetes should have the possibility of CRC in mind, and gastroenterologists that diagnose CRC or colorectal neoplasia, should think about the cardiovascular health of their patients. We, therefore, propose five distinct clinical scenarios, based on our results and pre-existing clinical evidence on how an updated, more individualized screening approach might look like, as outlined in Figure 3.

Epidemiological data of the potential dimension of the clinical problem of patients with CRC and high cardiovascular risk is sparse. We report, in a real-world screening population for CRC, 5.2% of subjects with CAD and, depending on the definition, approximately 15% of subjects with high cardiovascular risk. Taken together, approximately 20% of our screened subjects, that is about four hundred and twenty out of two thousand and ninety-eight subjects, may be defined as the scope of the clinical problem, with potential overlap between colorectal neoplasia and cardiovascular disease. With an estimated number of 14–15 million people with cardiovascular disease (excluding hypertension) in the United States alone, that also have a history of cancer, this number might even be higher [67,68]. If our findings apply to every fifth patient in a real-world screening population, an average cardiologist, gastroenterologist, or a diabetologist, will face the clinical scenario we report on (i.e., a subject with shared risk factors for CRC and CAD), virtually every day.

### 4.6. Strengths and Limitations

Although several studies on the topic of shared risk factors of CAD/CVR and CRC have been published previously, our investigation was unique for several reasons. First, no study so far assessed a Caucasian population with respect to cardiovascular risk and its relationship to CRC. CVD and CRC showed considerable ethnic differences, and studies that were mostly conducted in Asia do not necessarily reflect the Caucasian situation. Furthermore, exactly for the same reason, the HS was developed by the ESC (European Society of Cardiology), as existing risk-prediction tools, like the FRS only insufficiently predicted cardiovascular risk in Europe, and the HS of the ESC has two versions for low-risk and high-risk countries in Europe, to account for the regional differences [69,70,71]. No previous study used HS to correlate findings in a screening colonoscopy. In addition, we did not exclude subjects with known CAD, and showed that in known CAD, the risk for colorectal neoplasia is even higher than in the non-CAD population. The increase in OR per percentage point of both FRS and HS we reported, was in line with this finding. With respect to the assessment of diabetes and metabolic syndrome, we reported a detailed investigation of the metabolic profile, including oral glucose tolerance test in every patient.

In our study, CAD was only established via questionnaire or medical history. Coronary angiography was available in 51% of subjects that reported a known CAD and the angiographically-verified CAD patients did not differ significantly in any aspect, as compared to subjects that reported a known CAD, except for aspirin use. This finding and the fact that self-reported CAD is significantly associated with CVR and also risk for CRC, make a relevant wrongful reporting of known CAD in subjects with unknown coronary status, unlikely. However, some subjects might not be aware of the fact that they actually have CAD. In case our records did not reveal a known CAD, we might naturally have missed subjects with known CAD. We think such a misrepresentation would only marginally change our results. In addition, it would be highly unethical to screen such a large cohort using coronary angiography, the gold standard for the diagnosis of CAD. Additionally, a Hong Kong study used subjects that needed a coronary angiography due to suspected CAD, which also imposed a bias to the study population. Other screening methods for CAD in asymptomatic subjects were shown to be problematic, due to the low sensitivity and specificity, so we think that no substantial improvement of a definite diagnosis of CAD is feasible in such a large screening cohort [15,72]. Although we included a large number of subjects, the number of advanced colorectal neoplasia per site was too low to draw reliable conclusions. Our results should be regarded as rather hypothesis-generating, and future studies with even larger cohorts are needed to answer such site-specific questions.

## 5. Conclusions

In conclusion, we report a risk factor-dependent association between the cardiovascular risk factors and the incidence of neoplasia, in both the proximal and the distal colon, presumably due to the shared risk factors. The prevalence of adenomas in the rectum was much lower than in the colon, and significant incremental risk could be shown for the FRS but not the HS tertiles. We further report that subjects with type 2 diabetes had significantly more adenomas in the proximal than in the distal colon. This might highlight the importance of a complete colonoscopy, especially, in patients with type 2 diabetes. Our data suggest screening colonoscopy to be indicated particularly in subjects with known coronary artery disease or high cardiovascular risk, in order to detect potentially treatable colorectal neoplasia.

## Figures and Tables

**Figure 1 jcm-07-00484-f001:**
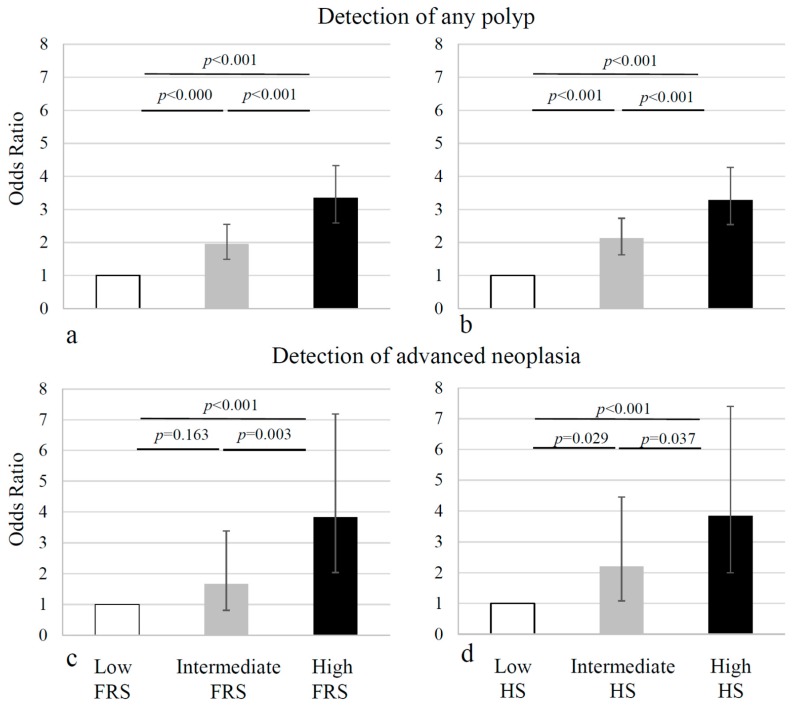
Odds ratios for any adenoma and advanced neoplasia. The odds ratios (ORs) for the prevalence of any adenoma, more than doubled in the Framingham Risk Score (FRS) and the Heart Score (HS) of the European Society of Cardiology intermediate tertiles, and tripled in the upper FRS and HS tertiles, relative to the respective low tertiles (**a**,**b**). ORs for having advanced neoplasia only marginally increased in the intermediate tertiles, but showed a more than three-fold increase in the upper tertiles of the FRS and the HS (**c**,**d**).

**Figure 2 jcm-07-00484-f002:**
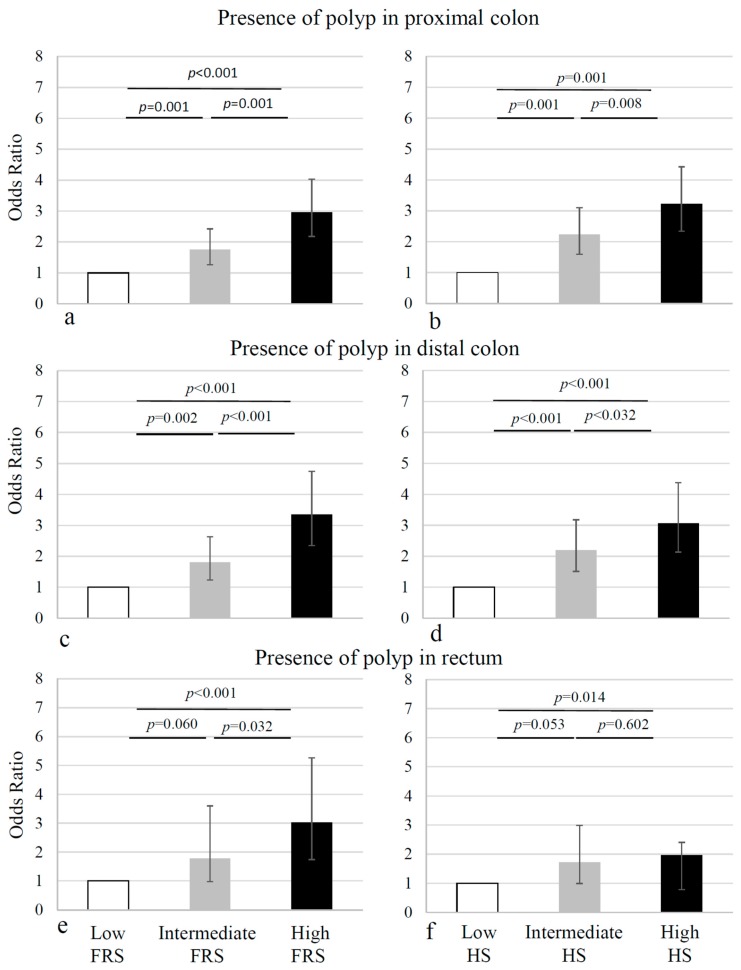
Site-specific odds ratios for any adenoma and advanced neoplasia. Odds Ratios (ORs) for having lesions in the proximal or distal colon increased in the intermediate and the upper Framingham Risk Score (FRS) and the Heart Score (HS) of the European Society of Cardiology tertiles, respectively (**a**–**d**). The prevalence of adenomas in the rectum was much lower than in the colon. The difference in the distribution of the rectal lesions, according to the cardiovascular risk, was more pronounced in the FRS (*p* < 0.001) than in the HS tertiles (*p* = 0.039) (**e**,**f**).

**Figure 3 jcm-07-00484-f003:**
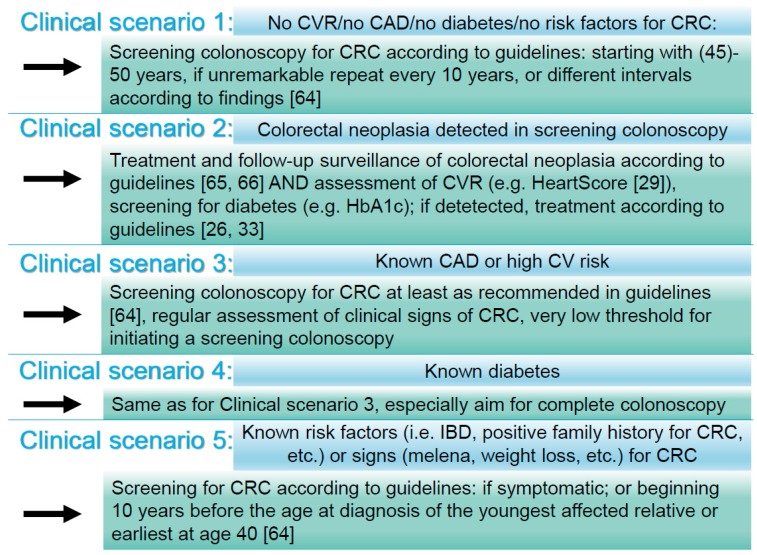
A proposal for an individualized screening colonoscopy approach for colorectal cancer, diabetes, cardiovascular risk, and coronary artery disease, based on our findings. Along with screening colonoscopy, other screening tests are likewise possible (fecal immunochemical test annually; high-sensitivity, annual guaiac-based fecal occult blood test; multitarget stool DNA test every three years; computed tomography colonography every five years; and flexible sigmoidoscopy every five years [26,29,33,64,65,66]). CVR denotes cardiovascular risk, CAD denotes coronary artery disease, CRC denotes colorectal cancer, and IBD denotes inflammatory bowel disease.

**Table 1 jcm-07-00484-t001:** Characteristics of subjects with and without CAD History.

Variable	No CAD History	CAD History	*p*
*N*	1990	108	
Age (years, [mean ± SD])	58.7 ± 9.7	66.6 ± 7.5	<0.001
Males (*n*, [%])	982 (49.4)	69 (63.9)	0.003
BMI (kg/m^2^)	27.1 ± 4.8	28.2 ± 4.5	0.026
Systolic blood pressure (mmHg)	133.7 ± 19.2	131.7 ± 18.6	0.280
Diastolic blood pressure (mmHg)	79.6 ± 9.0	80.9 ± 11.1	0.229
Smoking status (never/ever/current, *n*, [%])	968 (48.6)/703 (35.4)/319 (16.0)	49 (45.4)/46 (42.6)/13 (12)	0.250
Hypertension (*n*, [%])	1242 (62.4)	70 (64.8)	0.615
IFG (*n*, [%])	143 (7.3)	24 22.2	<0.001
IGT (*n*, [%])	330 (16.5)	21 (19.4)	0.437
HOMA-IR	2.8 ± 6.1	4.6 ± 6.5	0.003
HbA1c [%]	5.8 ± 0.7	6.1 ± 0.7	<0.001
Diabetes (*n*, [%])	262 (13.2)	35 (32.4)	<0.001
Metabolic Syndrome (*n*, [%])	385 (19.4)	39 (36.1)	<0.001
FRS	7.1 ± 6.0	11.5 ± 6.8	<0.001
HS	3.0 ± 3.6	4.8 ± 3.6	<0.001
Aspirin (*n*, [%])	264 (13.2)	83 (76.9)	<0.001

Subjects with and without CAD (Coronary artery disease) history differed significantly, with respect to age, gender, BMI (body mass index), presence of type 2 diabetes impaired fasting glucose (IFG), Homeostatic Model Assessment for Insulin Resistance (HOMA-IR), glycated hemoglobin A1c (HbA1c), presence of metabolic syndrome, Framingham Risk Score (FRS), Heart Score (HS) of the European Society of Cardiology, and Aspirin use, but did not differ in smoking status, blood pressure (mmHg, millimeter of mercury), impaired glucose tolerance (IGT), and presence of arterial hypertension.

**Table 2 jcm-07-00484-t002:** Colorectal lesion and advanced neoplasia by CAD history.

CAD History	Total *N*	Subjects with Any Adenoma *N* (%)	*N* to be Screened	OR (95% CI)	*p*-Value	Subjects with Advanced Neoplasia	*N* to be Screened	OR (95% CI)	*p*-Value
No	1990	526 (26.4)	3.8	Reference		75 (3.8)	26.5	Reference	
Yes	108	38 (35.2)	2.8	1.51 (1.10–2.27)	0.047	10 (9.6)	10.8	2.62 (1.31–5.20)	0.007

Subjects with CAD (coronary artery disease) history had a higher detection-rate of any adenoma and of advanced neoplasia, as compared to the subjects without a CAD history.

**Table 3 jcm-07-00484-t003:** Risk of any adenoma and advanced neoplasia, by one percentage point increases in the risk scores, in subjects without known CAD.

Risk Score	Any Adenoma	*p*-Value	Advanced Neoplasia	*p*-Value
FRS (% points)	1.07 (1.06–1.09)	<0.001	1.07 (1.04–1.12)	<0.001
HS (% points)	1.09 (1.07–1.14)	<0.001	1.09 (1.05–1.14	<0.001

The risk of any adenoma and advanced neoplasia increased significantly by one percentage point rise in the Framingham Risk Score (FRS) and the Heart Score (HS) of the European Society of Cardiology.

**Table 4 jcm-07-00484-t004:** Risk of any adenoma or advanced neoplasia by the FRS and the HS risk tertiles.

		Any Adenoma						Advanced Neoplasia					
	*N*	*N* (%)	*N* to be Screened	OR (95% CI)	Sensitivity	Specificity	*p*	*N* (%)	*N* to be Screened	OR (95% CI)	*p*	Sensitivity	Specificity
FRS	1990	526 (26.4)	3.8					75 (3.8)	26.5				
Low (0–3)	711	111 (15.6)	6.4	Reference				13 (1.8)	54.7	Reference			
Intermediate (4–8)	633	168 (26.5)	3.8	1.95 (1.49–2.55)			<0.001	19 (3.0)	33.3	1.66 (0.81–3.39)	0.163		
High (>8)	646	247 (38.2)	2.6	3.35 (2.59–4.33)	47%	73%	<0.001	43 (6.7)	15.0	3.83 (2.04–7.19)	<0.001	57%	69%
HS	1990	526 (26.4)	3.8					75 (3.8)	26.5				
Low (<1.077)	711	109 (15.3)	6.5	Reference				12 (1.7)	59.2	Reference			
Intermediate (1.077–3.192)	633	176 (27.8)	3.6	2.13 (1.63–2.78)			<0.001	23 (3.6)	27.5	2.20 (1.08–4.45)	0.029		
High (3.192)	646	241 (37.3)	2.7	3.29 (2.54–4.26)	46%	72%	<0.001	40 (6.2)	16.2	3.84 (2.00–7.40)	<0.001	53%	68%

Framingham Risk Score (FRS) and Heart Score (HS) of the European Society of Cardiology tertiles showed significant associations with any adenoma and advanced neoplasia. ORs (95% CI) for FRS high vs. intermediate (reference), for any adenoma—1.71 (1.35–2.17) (*p* = 0.000). ORs (95% CI), for FRS high vs. intermediate (reference) for advanced neoplasia—2.30 (1.33–4.00) (*p* = 0.003). ORs (95% CI) for HS high vs. intermediate (reference), for any adenoma—1.55 (1.22–1.96) (*p* = 0.000). ORs (95% CI) for HS high vs. intermediate (reference), for advanced neoplasia—1.75 (1.04–2.96) (0.037).

**Table 5 jcm-07-00484-t005:** Associations of the Framingham Risk (FRS) Tertiles with colorectal lesion, by location.

FRS-Tertiles	*N*	*N* (%) with Any Adenoma	*N* to be Screened	OR (95% CI)	*p*-Value	Sensitivity	Specificity
Proximal Colon	1990	325 (16.3)	6.1				
Low	711	69 (9.7)	10.3				
Intermediate	633	100 (15.8)	6.3	Reference	0.001		
High	646	156 (24.2)	4.1	1.75 (1.26–2.42)	<0.001	48%	71%
Distal Colon	1990	247 (12.4)	8.1	2.96 (2.18–4.03)			
Low	711	48 (6.8)	14.8	Reference			
Intermediate	633	73 (11.5)	8.7	1.80 (1.23–2.63)	0.002		
High	646	126 (19.5)	5.1	3.35 (2.35–4.76)	<0.001	51%	70%
Rectum	1990	93 (4.7)	21.4				
Low	711	18 (2.5)	39.5	Reference			
Intermediate	633	28 (4.4)	22.6	1.78 (0.98–3.25)	0.060		
High	646	47 (7.3)	13.7	3.02 (1.74–5.26)	<0.001	51%	68%

Framingham Risk Score (FRS) tertiles showed significant associations with detection-rates of any adenoma by location (proximal colon, distal colon, and rectum). ORs (95% CI) for FRS high vs. intermediate (reference), for presence of adenoma in prox. colon—1.70 (1.28–2.24) (*p* = 0.000). ORs (95% CI) for FRS high vs. intermediate (reference), for presence of adenoma in dist. colon—1.86 (1.36–2.54) (*p* = 0.000). ORs (95% CI) for FRS high vs. intermediate (reference), for presence of adenoma in the rectum—1.70 (1.05–2.74) (*p* = 0.032).

**Table 6 jcm-07-00484-t006:** Associations of the Heart Score (HS) Tertiles with colorectal lesion, by location.

HS-Tertiles	*N*	*N* (%) with Any Adenoma	*N* to be Screened	OR (95% CI)	*p*-Value	Sensitivity	Specificity
Proximal Colon	1990	325 (16.3)	6.1				
Low	711	62 (8.7)	11.5				
Intermediate	633	111 (17.5)	5.7	Reference	<0.001		
High	646	152 (23.5)	4.3	2.23 (1.60–3.10	<0.001	47%	70%
Distal Colon	1990	247 (12.4)	8.1	3.22 (2.34–4.43)			
Low	711	47 (6.6)	15.1				
Intermediate	633	85 (13.4)	7.4	Reference	<0.001		
High	646	115 (17.8)	5.6	2.19 (1.51–3.18)	<0.001	47%	70%
Rectum	1990	93 (4.7)	21.4	3.06 (2.14–4.38)			
Low	711	22 (3.1)	32.3				
Intermediate	633	33 (5.2)	19.2	Reference	0.053		
High	646	38 (5.9)	17	1.72 (0.99–2.99)	0.014	41%	68%

The Heart Score (HS) of the European Society of Cardiology tertiles showed significant associations with detection rates of any adenoma by location (proximal colon, distal colon, and rectum). ORs (95% CI) for HS high vs. intermediate (reference), for presence of adenoma in prox. colon—1.45 (1.10–1.90) (*p* = 0.008). ORs (95% CI) for HS high vs. intermediate (reference), for presence of adenoma in dist. colon—1.40 (1.03–1.89) (*p* = 0.032). ORs (95% CI) for HS high vs. intermediate (reference), for presence of adenoma in rectum—1.14 (0.70–1.84) (*p* = 0.602).

**Table 7 jcm-07-00484-t007:** Correlation matrix, the upper number indicating Spearman’s rho, the lower indicating the *p*-values.

	IFG [yes/no]	IGT [yes/no]	HOMA [index]	FRS [%]	HS [%]	Polyps [*n*]	Tubular Adenoma [yes/no]]	Advanced Neoplasia [yes/no]	Adenoma in Proximal Colon [*n*]	Adenoma in Distal Colon [*n*]	Adenoma in Rectum [*n*]
IFG [yes/no]	—										
	—										
IGT [yes/no]	0.123	—									
	<0.001	—									
HOMA [index]	0.283	0.105	—								
	<0.001	<0.001	—								
FRS [%]	0.176	0.15	0.25	—							
	<0.001	<0.001	<0.001	—							
HS [%]	0.184	0.14	0.208	0.802	—						
	<0.001	<0.001	<0.001	<0.001	—						
polyps [*n*]	0.031	0.054	0.111	0.238	0.237	—					
	0.16	0.013	<0.001	<0.001	<0.001	—					
tubular adenoma [yes/no]]	0.039	0.045	0.095	0.212	0.222	0.92	—				
	0.077	0.04	<0.001	<0.001	<0.001	<0.001	—				
advanced neoplasia [yes/no]	0.005	0.053	0.053	0.1	0.096	0.317	0.087	—			
	0.833	0.015	0.047	<0.001	<0.001	<0.001	<0.001	—			
adenoma in proximal colon [*n*]	0.02	0.043	0.07	0.179	0.191	0.728	0.707	0.223	—		
	0.36	0.047	0.008	<0.001	<0.001	<0.001	<0.001	<0.001	—		
adenoma in distal colon [*n*]	0.031	0.038	0.117	0.165	0.156	0.64	0.592	0.221	0.169	—	
	0.152	0.081	<0.001	<0.001	<0.001	<0.001	<0.001	<0.001	<0.001	—	
adenoma in rectum [*n*]	0.024	0.028	0.033	0.096	0.081	0.367	0.305	0.15	0.036	0.115	—
	0.264	0.194	0.216	<0.001	<0.001	<0.001	<0.001	<0.001	0.096	<0.001	—

**Table 8 jcm-07-00484-t008:** Sex-specific differences with regard to cardiovascular risk and location of neoplasia.

	Males	Females	*p*-Value
*n*	1051	1047	n.a.
Age (years; mean ± SD)	58.8 ± 9.4	59.5 ± 10.1	0.152
CAD (*n* [%])	69 (6.6)	39 (3.7)	0.003
FRH (%; mean ± SD)	10.6 ± 6.1	4.0 ± 4.1	<0.001
HS (%; mean ± SD)	3.7 ± 3.9	2.5 ± 3.3	<0.001
Arterial hypertension (*n* [%])	701 (66.7)	611 (58.4)	<0.001
BMI (kg/m^2^; mean ± SD)	27.6 ± 4.1	26.8 ± 5.5	<0.001
Diabetes mellitus (*n* [%])	161 (15.3)	136 (13.0)	0.126
Any adenoma	371 (35.3%)	218 (20.8%)	<0.001
Any adenoma (total number of adenomas detected) (*n* [%])	630	338	<0.001
Advanced neopasia (*n* [%])	67 (6.4)	41 (3.9)	0.011
Adenoma in proximal colon (*n* [%])	220 (20.9)	130 (12.4)	<0.001
Adenoma in distal colon (*n* [%])	166 (15.8)	100 (9.6)	<0.001
Adenoma in rectum (*n* [%])	71 (6.8)	30 (2.9)	<0.001

Males had significantly more cardiovascular risk factors than females and, accordingly, more lesions were found in males. However, no site-specific differences could be found.

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
