# Peer review of "Association between Cardiovascular Risk and Diabetes with Colorectal Neoplasia: A Site-Specific Analysis"

_jcm, 2018, doi:10.3390/jcm7120484_

Reviewer 1 Report

GENERAL COMMENTS

This study assessed the relationship between known coronary artery disease or risk for CAD and the prevalence of colorectal neoplasms in the screening of 2098 asymptomatic persons from the general population in Austria, therefore, this is a single-center study.

Colorectal carcinoma is a very pertinent health problem in the general population, especially in Western societies. On the other hand, coronary heart disease is an equally important (if not even more prominent) health challenge in modern industrialized societies. Just these facts alone justify all research efforts that are directed towards these problems in medicine.

Study found that males had higher cardiovascular risk than women and had more findings in colonoscopy compared to women (without any site-specific differences) whereas patients with known CAD and high cardiovascular risk had a significantly increased risk of colorectal carcinoma in both proximal and distal region whereas proximal colon lesions were significantly associated with type II diabetes mellitus.

Statistical analyses are appropriate, although the different presentation of data would be advised.

Relevant ethical/legal disclosures are present and appropriately reported in the manuscript. Informed consent was obtained from all participants and participation was voluntary.

References are generally appropriate and relevant regarding this topic, however, should be reformatted within the text.

 SPECIFIC COMMENTS

1.      Please amend references and put them in the parenthesis and at the end of each sentence, not within sentences as it is currently present. Fix and check all references in the reference list as well and make sure they adhere to Journal guidelines.

 Please refer to

https://www.mdpi.com/journal/jcm/instructions#preparation

 2.      In the Introduction authors state that colorectal carcinoma has its site-specific differences in terms of histology, clinical characteristics etc. however, this is not substantiated with evidence. In the line, 52/53 authors state this as a fact, however, do not provide references to these „more recent studies demonstrating“...
Please substantiate this claim with appropriate references.

3.      Please elaborate more on why site-specific differentiation of colorectal carcinoma is clinically relevant at all. An average reader might not know exactly how does it impact on prognosis and patient outcomes? This needs to be elaborated more thoroughly, especially in the context that site-specific analysis is one of the main analysis outcomes in this paper.

4.      This was a single-center study conducted at Hospital in Austria. How do cardiovascular risk profile and its connection to colorectal disease vary among a different population, ethnic/continental etc.? Do we have varying data on this among European population vs. Asian, American, etc.? It would be beneficial to discuss this in the Discussion section.

5.      Framingham Coronary Heart Disease Risk score is intended for a population of 30-74 years and included diabetes mellitus, however, this is an older risk stratification model. Authors used one by Wilson et al. Circulation 1998. However, since in the Methods you state that you excluded subjects >79 years, that implies that you have been using an updated Framingham risk model, however, that one cannot include patients with T2DM. In the references, you refer to Wilson et al. 1998 which is an older version of Framingham risk score and that one has a cut-off of 74 years, in that case, you should have excluded patients older than 74.
Please state clearly which Framingham risk model was used and cite it with appropriate references and explain which patients were excluded and why?
These days Type 2 DM is considered as a CHD risk equivalent. You might have to exclude patients with Diabetes in FRS.

6.      Regarding the HeartScore model, please make sure that you used the European LOW-risk model that is suitable for the Austrian population, and if yes, make sure that you define this in the revised version.

7.      Additionally, HeartScore is validated in a population of 40-65 and does not recommend diabetic patients to be included in risk calculation?
Did you include diabetic patients in this calculation? Alternatively, diabetic patients should be a view, by the default, as high-risk patients. Please define this.

8.      Please report odds ratios (ORs) with respect to FRS and HS tertiles in the table rather than in the Figure. Specifically, this refers to data shown in Figure 1. and Figure 2. Also, report 95% confidence interval and p-values.

9.      While you do a report on logistic regression analysis as a univariate, meaning that risk score was plotted against the neoplasm outcome of interest, this statistical approach might not be fully suitable. Framingham risk score incorporates age, sex, smoking status, total cholesterol, HDL cholesterol, and SBP, etc. as variables in predicting 10-year risk of MI or death. On another hand, HS includes similar variables, meaning age, sex, SBP, total/HDL cholesterol and smoking status for the estimation of 10-year risk of mortality due to MI, stroke or circulatory problem. Could the authors set these variables as independent variables in the model and run it against positive adenoma status in different regions set as the dependent variable?

10.  I suggest determining AUC through the ROC analysis for the high-risk scores obtained on FRS and HS in terms of sensitivity and specificity for the presence of adenoma in distal, proximal colon and rectum.

11.  In the discussion, authors should discuss what could be possible pathophysiological reasons to why proximal and distal colon and not rectum were associated with high scores on FRS and HS, in terms of cardiovascular risk factors and what practical implications might that have? How does this impact on practice and clinical/diagnostic/treatment workup of these patients. You emphasize the point that complete colonoscopy might be a prudent idea in T2DM patients. What is the logic for patients without T2DM?

12.  This is an interesting paper, however, site-specific differentiation of colorectal neoplasms does not have more than academical meaning if this differentiation does not imply on certain amendments in clinical practice or has some sort of clinical/prognostic utility. I believe authors should dedicate more discussion points to elaborate on these aspects of this research, of course, in the light of the currently available evidence, which is not substantial at this point, as authors state.

Author Response

The point-by point response to the comments of reviewer #1 are given in the attached file.

Reviewer 2 Report

David Niederseer et al. has written and presented the data clearly and fits the scope of this journal. This is an exciting paper discussing about the link between CVD patients and Colorectal cancer patients. Initially, I was quite surprised to these this paper, as the author published a similar kind of study in JACC 2017. But the author clearly explained the differences between the two study, They have also used HS score to validate their claim. 

Can the author confirm that they did not use the same set of patients sample for both studies?

It would be more informative if the author can provide the data on heart function and blood pressure.

  In their JACC paper, the author has mentioned "FRS does not include diabetes, obesity, red meat or saturated fat intake or family history of colorectal cancer. All these parameters are relevant for the assessment of colorectal cancer risk". To address part of this issue, Author has included HS score in this study. It would be interesting if the author can explain how FRS and Colorectal cancer can be linked based on the above statement.

In the title the author has mentioned an association between Cardiovascular risk factors and colorectal neoplasia," but in the article, the author has also addressed the link between diabetes and colorectal cancer. Even in conclusion mentioned diabetes. The author should add diabetes to the title.

As the Author was discussing about diabetes, Author should provide some details about patients glucose levels, impaired glucose tolerance and also insulin resistant. All these parameters are mostly linked with colorectal cancer.

Author Response

The point-by-point respose to the comments of reviewer #2 are attached in the file.

Round  2

Reviewer 1 Report

Authors have addressed all my concerns.

Author Response

We thank Reviewer #1 for the comments in the first round of the review process. We certainly feel that because of these comments the quality of the manuscript improved substantially. We are glad that we could address all the concerns raised by Reviewer #1 in our point-to-point reply.